**Data Availability Statement:** Due to the personal data collected and potentially identifying information contained within the data, data are

# Maternal serum vitamin D level in early pregnancy and risk for preeclampsia: A case-control study in Southern Sweden

Gunilla Malm[1]*, Christian H. Lindh[1], Stefan R. Hansson[2], Karin Källén[1,3], Johan Malm[4], Lars Rylander[1]

1 Division of Occupational and Environmental Medicine, Lund University, Lund, Sweden, 2 Department of Obstetrics and Gynecology, Institute of Clinical Sciences Lund, Lund University, Lund, Sweden, 3 Tornblad Institute, Lund University, Lund, Sweden, 4 Department of Translational Medicine, Lund University, Lund, Sweden

☯ These authors contributed equally to this work.
* gunilla.malm@med.lu.se

## Abstract

### Background

Preeclampsia is considered a major cause of maternal and fetal morbidity and mortality. The aim of the present case-control study in Sweden was to assess the hypothesized association between low serum vitamin D concentrations in early pregnancy and the risk of developing preeclampsia since vitamin D may play a role in early placental development.

### Methods

The study included 296 women diagnosed with preeclampsia (cases) and 580 healthy pregnant women (controls). Serum samples were obtained from a biobank of samples collected in early pregnancy including almost all pregnancies in Southern Sweden. Concentrations of 25-hydroxyvitamin D3 (vitamin D) were analyzed using liquid chromatography-tandem-mass-spectrometry (LC/MS/MS). The cases were divided into two categories: i) infants were born before gestational week 34 (early onset) and/or born small-for-gestational age (SGA)(n = 51), ii) and others defined as late onset (n = 245). Vitamin D concentrations were analyzed both as a continuous and a categorized variable.

### Results

When all preeclampsia cases were included in the analyses no consistent patterns were observed. However, the median serum concentrations of vitamin D were significantly lower among the cases who were early onset and/or were born SGA (median 39.2 nmol/L, range 1.2–93.6) as compared to the controls (49.0 nmol/L, 0.1–219; p = 0.01). In addition, high concentrations were statistically significantly associated with a decreased risk of preeclampsia (>66.9 vs ≤30.1 nmol/L; crude OR 0.39, 95% CI 0.16–0.96). When potential confounders were included in the models the associations were even more pronounced.

available upon request. Ethical approval by The Swedish Ethical Review Authority may be necessary, and requests for data may be sent to LUPOP - Lund University Population Research Platform via email (lupop@ed.lu.se).

**Funding:** The work was supported by the Swedish Research Council for Sustainable Development (FORMAS, Dnr 217-00699), the ReproUnion, which was co-financed by the European Union, Interreg VÖKS, an ALF government funding (Dnr 2018-0075), and Lund University. The funders had no role in study design, data collection and analysis, decision to publish, or preparation of the manuscript. There was no additional external funding received for this study.

**Competing interests:** The authors have declared that no competing interests exist.

## Conclusions

Our results support the hypothesis that vitamin D deficiency is a risk factor for preeclampsia, but only in preeclampsia cases who were early-onset and/or were born SGA. Preeclampsia is not a homogenous condition and more studies are needed before vitamin D supplementation during pregnancy can be recommended.

## Background

Preeclampsia is considered a major cause of maternal and fetal morbidity and mortality in developing countries [1]. The incidence of preeclampsia has increased in the past two decades [2] and is now seen in 2–8% of all pregnancies worldwide [3]. In Sweden, approximately 3% of pregnancies are complicated by preeclampsia [4].

Preeclampsia has for many years been defined as new onset hypertension and proteinuria developing in the second half of pregnancy and resolving after delivery. Refined definitions have been suggested, all including new onset hypertension. However, the most recent definition of preeclampsia, endorsed by the International Society for the Study of Hypertension in Pregnancy (ISSHP), describes the condition as a combination of new onset hypertension (systolic >140 mmHg and diastolic >90 mmHg) and one or more features of maternal organ dysfunction (e.g. liver, kidney, neurological), hematological involvement, uteroplacental dysfunction, fetal growth restriction, and abnormal Doppler ultrasound findings of uteroplacental blood flow [5].

Preeclampsia is a systemic disease affecting the vascular endothelium with an increased risk of future cardiovascular disease [6]. The epidemiology, clinical presentation and associated morbidity are very heterogenous and the pathobiological mechanisms causing the condition are far from fully understood. Early-onset or 'placental' preeclampsia (occurring before 34 weeks) is associated with substantial risk of intrauterine growth restriction, whereas late-onset or 'maternal' preeclampsia (occurring after 34 weeks) often is associated with maternal risk factors such as obesity and systemic diseases. The neonates are often large-for their gestational-age [5]. Early onset preeclampsia is a consequence of defective placentation and shares a common initiating pathophysiology to other disorders of placentation, especially fetal growth restriction (FGR), characterized by a reduced placental blood flow, oxidative stress and inflammation [7]. In contrast, late-onset preeclampsia is associated with maternal metabolic risk-factors causing placental stress [5, 7].

A better understanding of the pathogenesis of preeclampsia will help in preventing and treating the disease [8]. Recent studies suggest that low vitamin D levels in pregnancy are associated with increased risk of preeclampsia. Vitamin D may play a role in early placental development thus there might be a preventive role for vitamin D supplementation [9, 10].

However, the results concerning the association between vitamin D and preeclampsia are inconsistent and must be clarified [11].

Vitamin D deficiency is a global health problem and common in pregnant women, particularly in many low- and middle-income countries (LMIC) [12]. The prevalence of deficiency varies between different population groups and is much higher in women born in Africa and Asia than in women born in the Western world [4]. According to the Swedish food agency, one fourth to one third of pregnant women in Sweden have vitamin D levels <50 nmol/L and about 10% <30 nmol/L (Swedish food agency, 2018). Factors that influence the vitamin D

status include ethnicity, geographical living latitude (i.e. sun exposure), season, sun-seeking behavior, clothing style, dietary vitamin D, sunscreen [2, 4].

To clarify the link between vitamin D status and preeclampsia, large population-based studies, with high participation rate as well as valid data regarding pregnancy outcomes and potential confounders are needed. In the present study, the association between vitamin D levels and the risk of developing preeclampsia was investigated in a population-based study by combining comprehensive information from registers and concentrations of vitamin D in maternal serum collected in early pregnancy before manifestation of preeclampsia.

## Material and methods

### Study design and sources

This case-control study was performed in the most Southern County of Sweden, Scania (Skåne) and performed in accordance with the Declaration of Helsinki. The study was approved by the Ethical committee at Lund University, Sweden with the approval number 2014/696. The ethical approval included that a specific consent for the study was not needed.

The population comprised pregnant women during the period 1995–2009 obtained from the general population attending antenatal care clinics. Visits to antenatal care clinics are free of charge and therefore basically all pregnant women attend a standardized antenatal care program. The sample collection was performed in the first trimester (12–14 gestational weeks, gw) of pregnancy according to the standard regime for pregnant women in Sweden.

### The following sources were used

*Biobank*: Serum samples were obtained from the Southern Sweden Maternal Biobank (SSMB), a biobank in Scania, started in 1989, containing >250.000 serum samples collected in early pregnancy taken in connection with screening for infections and German measles (rubella) [13].

*Register data*: The Swedish Medical Birth Registry (MBR) includes virtually all children born in Sweden in 1973 or later [14]. The register was used in combination with a local birth register for Scania, Perinatal Revision South [15]. In addition, as part of the regional Scania ultrasound routines, examinations in early (18gw) and late pregnancy (32 gw), have been performed and registered in a separate database [16, 17].

All cases and controls were identified from the birth register and the ultrasound database. Conditions and diagnoses are recorded using checkboxes and/or with International Classification of Diseases code (ICD), where the 9[th] revision was used before 1998 and the 10[th] revision from 1998 and after. Preeclampsia was identified through marked checkboxes for moderate or severe preeclampsia, or by the presence of ICD-codes 642E, 642F (ICD9), or O140, O141, or O149 (ICD10), respectively. The controls were randomly chosen among women who were not diagnosed with preeclampsia and whose children were appropriate for gestational age (AGA). Women with an SGA child were excluded since these controls were included in an on-going SGA study.

Out of 450 preeclampsia cases and 900 randomly selected controls, 93% (n = 420) and 95% (n = 854), respectively, were available in the biobank. The cases and controls were randomly sorted, and the first 304 preeclampsia cases and 603 controls were selected. A priori power calculation indicated that 300 cases were needed to show a relevant statistical difference (Rylander et al., 2020) [18]. Due to small sample volumes, it was not possible to analyze vitamin D concentrations in eight preeclampsia cases and 23 controls. The final number of samples included were 296 preeclampsia cases and 580 controls.

## Analysis of vitamin D

Quantitative analyses of 25-hydroxyvitamin D3 were performed with liquid chromatography-tandem-mass-spectrometry (QTRAP 5500; AB Sciex, Framingham, MA, USA), at the division of Occupational and Environmental Medicine in Lund, Sweden using a method previously described by Gustafsson et al. 2015 [19]. In brief, aliquots of 75 μl serum were added with labelled internal standard $D_6$-25-hydroxyvitamin $D_3$. The proteins were precipitated with 200 μl acetonitrile and vigorously shaken for 30 min at room temperature. The samples were prepared and analysed in random order. In all analytical batches a total of four quality controls (QC) were done per sample batch, two prepared in- house and two QC obtained from Chrom-systems Instruments & Chemicals GmbH (MassCheck®; Gräfelfing, Germany). The QC samples were within specified range and the coefficient of variation (CV) was between 8–13%, similar to what has been reported in other studies, e. g. [20].

## Other pregnancy information

Additional information was collected in early pregnancy from the MBR including maternal body mass index (BMI, kg/m$^2$), parity, maternal smoking (three categories: non-smokers, 1–9 cig/day, and >9 cig/day), diabetes (yes/no), gestational diabetes (yes/no), and maternal country of origin (8 different categories, see Table 1). The exact date when the serum samples were collected was not available. However, based on information about date of childbirth and gestational length and the assumption that the serum samples were collected in gestational week 12, we could estimate which month of the year the sample was taken.

## Statistics

Initially, comparisons of serum concentrations of vitamin D in early pregnancy as a continuous variable between cases and controls was tested by Mann-Whitney test. In addition to treat the cases as one group, the cases were divided into two subgroups: i) those whose children were born before gestational week 34 (early onset) and/or born small-for-gestational age (SGA)(n = 51), ii) others, defined as 'late onset' (n = 245). We performed also separate comparisons among i) primipara women, ii) women whose country of origin were Sweden or the other Nordic countries. This differentiation was done due to the fact that previous studies have shown associations between vitamin D and preeclampsia for primipara [21] and that differences concerning vitamin D levels have been seen between different ethnic groups in Swedish studies [22, 23].

In a second step, logistic regression was applied which generated odds ratios (ORs) and 95% confidence intervals (CIs). In these analyses, the concentrations of vitamin D were categorized into quartiles, which were based on the distributions among the controls (30, 49, and 67 nmol/L), or predefined cut-offs (25, 50, and 75 nmol/L) based on levels suggested in the literature [24]. The lowest exposure quartile was used as the reference category. The crude analyses were completed with a priori defined models adjusted for maternal age (4 categories: <25, 26–30, 31–35, and >35 years), BMI (4 categories: <20, 20-<25, 25-<30, ≥30 kg/m$^2$), smoking (2 categories: yes and no), and parity (2 categories: 1 and ≥2). In addition, we did adjust for the estimated month of the year when the sample was taken. The other background variables in the present study had previously been shown to either no be statistically significantly associated with preeclampsia or judged to have categories with too few individuals (Rylander et al., 2020) [18]. However, it is important to stress that in some of the adjusted analyses the cases were relatively few, and these results must be interpreted with cautions. In line with this, we did not perform separate logistic regression analyses among primipara women and women whose country of origin were Sweden or the other Nordic countries.

**Table 1. Background characteristics among 296 preeclampsia cases and 580 controls.** In addition, the cases are divided into two groups. Numbers and within brackets median levels of vitamin D (25-hydroxyvitamin D3, nmol/L) are shown[b].

| | Early onset[a] and/or SGA (n = 51) | Cases<br>Other (n = 245) | Total (n = 296) | Controls<br>(n = 580) |
|---|---|---|---|---|
| | N (median[b]) | N (median[b]) | N (median[b]) | N (median[b]) |
| Maternal age at pregnancy (yr) | | | | |
| ≤20 | 4 (-) | 4 (-) | 8 (9.9) | 17 (37.1) |
| 21–25 | 9 (30,5) | 45 (36.2) | 54 (36.1) | 127 (43.5) |
| 26–30 | 16 (43.6) | 87 (60.2) | 103 (59.8) | 205 (49.6) |
| 31–35 | 15 (39.9) | 70 (56.6) | 85 (55.9) | 157 (52.3) |
| >35 | 7 (21.4) | 39 (49.3) | 46 (47.8) | 74 (50.2) |
| BMI in early pregnancy (kg/m²) | | | | |
| <20 | 4 (-) | 17 (52.5) | 21 (46.2) | 60 (51.3) |
| 20-<25 | 21 (36.2) | 111 (54.6) | 132 (51.2) | 286 (50.6) |
| 25-<30 | 9 (39.9) | 43 (56.6) | 52 (51.2) | 109 (46.2) |
| ≥30 | 6 (38.8) | 38 (50.5) | 44 (48.9) | 48 (40.4) |
| Missing | 11 | 36 | 47 | 77 |
| Parity | | | | |
| 1 | 41 (37.6) | 187 (52.4) | 228 (49.9) | 273 (51.2) |
| ≥2 | 10 (39.5) | 58 (59.3) | 68 (56.0) | 307 (47.0) |
| Gestational age (weeks) | | | | |
| -34 | 13 (39.9) | 0 | 13 (39.9) | 2 (-) |
| 35–37 | 24 (32.2) | 65 (49.3) | 89 (45.1) | 13 (44.0) |
| >37 | 14 (39.1) | 180 (55.6) | 194 (54.4) | 565 (49.1) |
| Small for Gestational Age (SGA) | | | | |
| Yes | 41 | 0 | 41 | 0 |
| No | 10 | 245 | 255 | 580 |
| Smoking in early pregnancy | | | | |
| Non smokers | 34 (36.2) | 206 (53.7) | 240 (51.2) | 490 (49.0) |
| 1–9 cig/day | 6 (37.9) | 6 (63.8) | 12 (45.8) | 39 (55.0) |
| >9 cig/day | 1 (-) | 3 (-) | 4 (-) | 13 (46.4) |
| Missing | 10 | 30 | 40 | 38 |
| Gender | | | | |
| Boys | 22 (36.2) | 131 (51.9) | 153 (49.9) | 299 (51.1) |
| Girls | 29 (42.3) | 114 (55.6) | 143 (52.4) | 281 (47.2) |
| Maternal country of origin | | | | |
| Sweden | 25 (46.4) | 166 (57.3) | 191 (56.6) | 369 (57.8) |
| Other Nordic countries | 1 (-) | 9 (46.9) | 10 (43.4) | 20 (54.4) |
| Western Europe, USA, Australia, New Zeeland | 0 (-) | 3 (-) | 3 (-) | 6 (49.6) |
| Previous Eastern Europe | 9 (21.3) | 15 (50.8) | 24 (37.5) | 48 (34.2) |
| Sub-Saharan Africa | 1 (-) | 5 (36.2) | 6 (30.3) | 13 (16.7) |
| Middle East | 5 (8.0) | 22 (13.4) | 27 (12.1) | 76 (10.7) |
| East Asia | 3 (-) | 2 (-) | 5 (36.7) | 23 (33.5) |
| South America | 0 (-) | 3 (-) | 3 (-) | 6 (38.4) |
| Missing | 7 | 20 | 27 | 19 |

[a] Early onset is defined as gestational weeks less than 34.

[b] Median values are not shown for groups with less than 5 individuals.

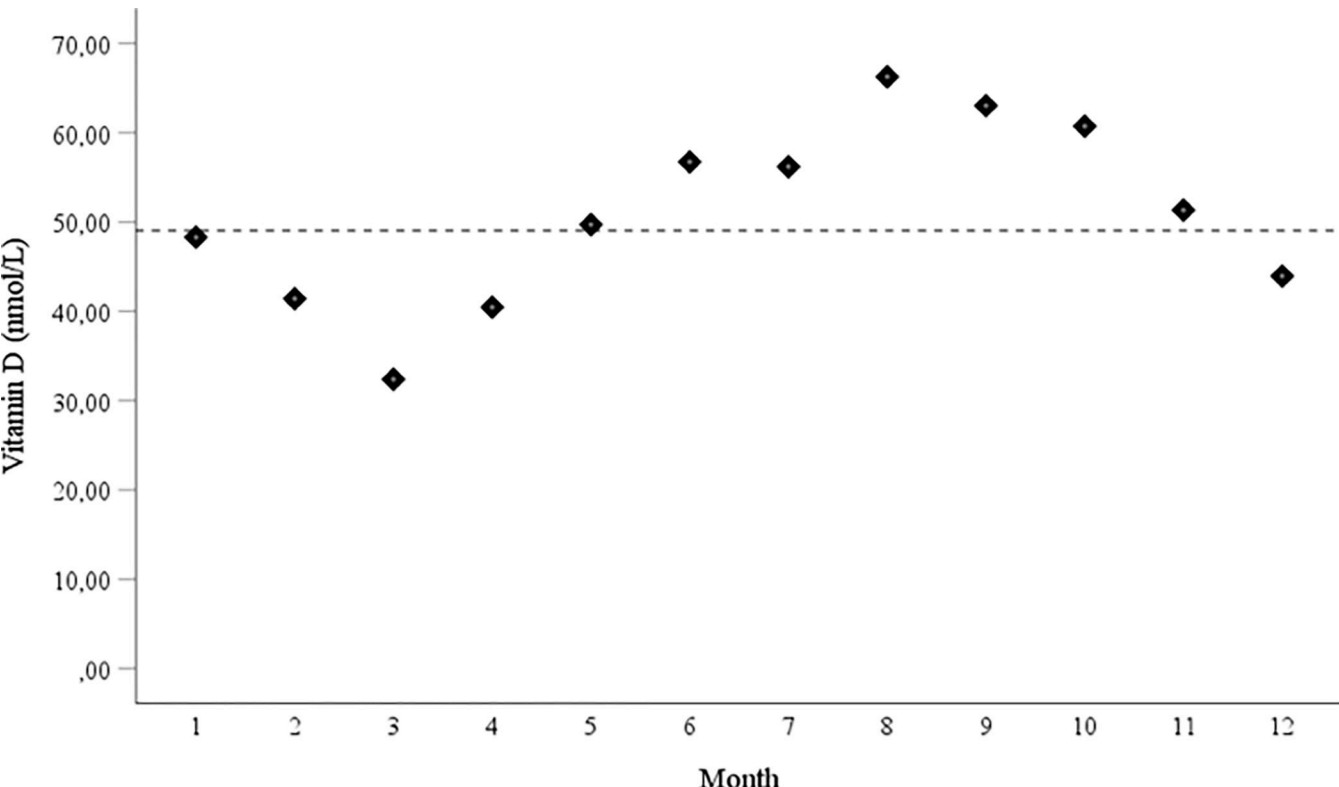

**Fig 1. Median concentrations of vitamin D (25-hydroxyvitamin D3) divided after estimated month of sampling in early pregnancy among the controls.** The number of controls per month varied between 40 and 64. The median for the whole period is also shown.

Statistical significance was defined as p-values below 0.05 or CIs not including 1.00. All analyses were performed with SPSS (version 27).

## Results

Background characteristics are shown in Table 1. As described previously, there was a significantly higher fraction of primipara and obese women among the cases as compared to the controls (Rylander et al., 2020) [18]. In Fig 1, the seasonal variation of vitamin D is shown among the controls. The lowest median concentration was observed in March whereas the highest median concentrations were observed during the late summer months.

The median serum concentration of vitamin D among all cases was 51.2 nmol/L (range 0.9, 163) and among the controls 49.0 nmol/L (range 0.1, 219). There was no statistically significant difference between the distributions (p = 0.42, Table 2). The cases who were early onset and/or whose children were born SGA had significantly lower concentration of vitamin D compared to the controls, irrespectively of subgrouping. The late-onset preeclampsia cases tended to have higher concentrations compared to the controls (p = 0.053), but this tendency was not present if we only included primiparous (p = 0.91) or those with a Nordic Country of origin (p = 0.65), respectively.

When the concentrations of vitamin D were categorized into quartiles, based on the distributions among the controls, there were no significant associations when all cases were included in the analysis (Table 3). However, when only early-onset cases and/or cases whose children were born SGA were included the individuals in the highest quartile showed significantly decreased risk for preeclampsia (unadjusted OR 0.39, 95% CI 0.16–0.96). This

**Table 2. Median levels with range of vitamin D (25-hydroxyvitamin D3, nmol/L) among preeclampsia cases and controls.** In addition, p-values from the comparisons with the controls are shown.

| | Cases | | | Controls |
|---|---|---|---|---|
| | Early onset[a] and/or SGA | Other | Total | |
| | Median | Median | Median | Median |
| | (range) | (range) | (range) | (range) |
| | p-value[b] | p-value[b] | p-value[b] | |
| All | 39.2 | 53.7 | 51.2 | 49.0 |
| | (1.2–93.6) | (0.9–163) | (0.9–163) | (0.1–219) |
| | **0.01** | 0.053 | 0.42 | |
| Parity = 1 | 37.6 | 52.4 | 49.9 | 51.2 |
| | (1.2–93.6) | (3.2–136) | (1.2–136) | (0.9–219) |
| | **0.007** | 0.91 | 0.41 | |
| Maternal country of origin = Sweden or other Nordic countries | 46.1 | 56.8 | 56.4 | 57.8 |
| | (23.5–93.6) | (13.5–163) | (13.5–163) | (0.9–136) |
| | **0.03** | 0.65 | 0.81 | |

[a] Early onset is defined as gestational weeks less than 34.

[b] Comparisons with the controls (p-values from Mann-Whitney test).

association was more pronounced when we adjusted for potential confounders. When only late-onset cases were included in the analysis, the individuals in the third quartile showed significantly increased risk for preeclampsia in the unadjusted analysis, but the statistical

**Table 3. The associations between concentrations of vitamin D (25-hydroxyvitamin D3) in maternal serum in early pregnancy and preeclampsia.** The cut-offs were based on quartiles among the controls. Odds ratios (OR) and 95% confidence intervals (CI) were obtained from logistic regressions.

| | | | Unadjusted | | Adjusted[a, b] | |
|---|---|---|---|---|---|---|
| Vitamin D (nmol/L) | Cases (n) | Controls (n) | OR | 95% CI | OR | 95% CI |
| | All cases included | | | | | |
| ≤30.1 | 63 | 145 | 1.00 | - | 1.00 | |
| >30.1–49.0 | 74 | 145 | 1.18 | 0.78–1.76 | 0.97 | 0.60–1.56 |
| >49.0–66.9 | 85 | 145 | 1.35 | 0.90–2.01 | 1.04 | 0.64–1.68 |
| >66.9 | 74 | 145 | 1.18 | 0.78–1.76 | 0.99 | 0.61–1.61 |
| | Early onset[c] and/or SGA cases included | | | | | |
| ≤30.1 | 18 | 145 | 1.00 | - | 1.00 | |
| >30.1–49.0 | 16 | 145 | 0.89 | 0.44–1.81 | 0.67 | 0.29–1.54 |
| >49.0–66.9 | 10 | 145 | 0.56 | 0.25–1.24 | 0.43 | 0.17–1.10 |
| >66.9 | 7 | 145 | **0.39** | **0.16–0.96** | **0.10** | **0.02–0.48** |
| | Other cases included | | | | | |
| ≤30.1 | 45 | 145 | 1.00 | - | 1.00 | |
| >30.1–49.0 | 58 | 145 | 1.29 | 0.82–2.03 | 1.06 | 0.63–1.80 |
| >49.0–66.9 | 75 | 145 | **1.67** | **1.08–2.58** | 1.28 | 0.76–2.16 |
| >66.9 | 67 | 145 | 1.49 | 0.96–2.32 | 1.37 | 0.81–2.30 |

[a] Adjusted for maternal age (4 categories: ≤25, 26–30, 31–35, and >35 years), BMI in early pregnancy (4 categories: <20, 20-<25, 25-<30, ≥30 kg/m$^2$), maternal smoking in early pregnancy (2 categories: yes and no), and parity (2 categories: 1 and ≥2).

[b] Include only individuals with complete data.

[c] Early onset is defined as gestational weeks less than 34.

significance was lost in the adjusted analysis. The pre-defined cut-offs were similar to the ones based on the distributions among the controls and the effect estimates were accordingly similar.

## Discussion

Overall, there was no significant association between serum levels of vitamin D in early pregnancy and the subsequent risk of developing preeclampsia when all cases were included in the analysis. However, low vitamin D levels were negatively associated when only early-onset cases and/or cases whose children were born SGA were included in the analysis. In addition, to evaluate robustness of this significant association, subgroup analyses were performed when vitamin D level was analyzed as a continuous variable. The results for these subgroup analyses did not differ from the results for the entire cohort. Since the study was performed in southern Sweden (latitude 56˚N) the results could be different at other latitudes, e.g. in northern Sweden (latitude 69˚N), north of the Arctic circle.

The present results stress the importance of dividing preeclampsia into different categories which has not been done in most of the previous studies, although Robinson and colleagues presented similar results already in 2010 [25]. Regarding total preeclampsia, Bodnar et al. reported a five-fold increased risk for preeclampsia in 55 women with vitamin D levels <37,5 nmol/L at <22 weeks of pregnancy [21]. Achkar et al. noted significantly lower vitamin D levels (<30 nmol/L) in early pregnancy (14 gw) among 169 women who later developed preeclampsia [26]. On the contrary, Benachi et al. found no association between vitamin D insufficiency (50 nmol/L) in the first trimester and the risk of developing preeclampsia (83 cases) [27].

As expected, there were seasonal variations in vitamin D with lower levels during the dark period of the year and the opposite during the lighter part of the year. The study samples were collected during all four seasons but the results did not differ when the sampling month was considered.

The protective effect of vitamin D on placental vascularization and angiogenesis takes place during placentation in early pregnancy [28]. It is therefore important to have adequate antenatal levels of vitamin D plasma levels. The mechanism behind the association between vitamin D and placenta function has not been fully elucidated. Calcitriol has been shown to regulate immune response and maintain immune homeostasis and thereby preventing placental vasoconstriction and ultimately preeclampsia. Also, effects on endothelial and smooth muscle cell proliferation and blood pressure regulation have been suggested to be involved in the interaction between vitamin D and cells in the placenta, possibly via vascular endothelial growth factor (VEGF)-mediated effects on angiogenesis [11].

The present study has several strengths. First, the study is based on a relatively large number of women diagnosed with preeclampsia which makes possible analyses also when diving into subtypes of preeclampsia. Second, the diagnoses were made by obstetricians and reported to a national register in a standardized way. Third, all serum samples were collected in early pregnancy, several weeks before manifestations of symptoms and before diagnosis. Forth, the vitamin D concentrations were determined using state-of-the-art analytical methodology. Although some of the serum samples have been stored for many years, the concentrations follow the expected yearly pattern with low winter levels and higher summer levels. Fifth, the registers contained several potential confounders and/or effect modifiers.

The study does also have some limitations. The major weakness in this study is that there is only one measurement of vitamin D per woman. In addition, the sample collections took place according to the national program for pregnant women, at the antenatal care clinics. Despite

this standardized routine the blood samples may have been handled differently at the various antenatal clinics. The serum samples have been stored at -20 or-80 degree Celsius during the period 1995–2009. A variation in the temperature may have affected the results but we consider this highly unlikely.

Which vitamin D level that should be considered as deficiency is still under debate [2]. Kiely et al. observed an independent protective association between serum vitamin D concentrations >75 nmol/L and reduced risk of preeclampsia with SGA [29]. The European Food Safety Authority (EFSA) and the British Scientific Advisory Committee on Nutrition (SACN) concluded that there is no evidence that higher vitamin D status is required later in pregnancy and during lactation. Therefore the dietary reference value for pregnant and lactating women equals that of women in general (Swedish food agency, 2018). However, it is still possible that non-Scandinavian women might benefit from a pre-pregnancy control to ensure normal levels of vitamin D before becoming pregnant or to measure it in the first trimester of pregnancy [30].

## Conclusions

In conclusion, low vitamin D levels were associated with preeclampsia among women with early-onset preeclampsia and/or whose children were born SGA. Preeclampsia is not a homogenous condition and more studies are needed before vitamin D supplementation during pregnancy can be recommended. The results stress the importance that future studies, epidemiological studies as well as randomized trials, should not treat preeclampsia as a homogeneous pregnancy complication.

## Acknowledgments

A special thank you to the laboratory staff, Åsa Amilon, Agneta Kristensen and Margareta Maxe at the Division for Occupational and Environmental Medicine at Lund University for excellent contribution in the laboratory.

## Author Contributions

**Conceptualization:** Stefan R. Hansson, Karin Källén, Lars Rylander.

**Formal analysis:** Gunilla Malm, Christian H. Lindh, Lars Rylander.

**Funding acquisition:** Stefan R. Hansson, Lars Rylander.

**Investigation:** Stefan R. Hansson, Karin Källén, Lars Rylander.

**Methodology:** Gunilla Malm, Christian H. Lindh, Stefan R. Hansson, Karin Källén, Lars Rylander.

**Project administration:** Stefan R. Hansson, Lars Rylander.

**Resources:** Lars Rylander.

**Supervision:** Lars Rylander.

**Validation:** Stefan R. Hansson.

**Writing – original draft:** Gunilla Malm.

**Writing – review & editing:** Gunilla Malm, Christian H. Lindh, Stefan R. Hansson, Karin Källén, Johan Malm, Lars Rylander.

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
