## [Decision Letter · Decision Letter 0]

11 Aug 2022

PONE-D-22-04216Maternal serum vitamin D level in early pregnancy and risk for preeclampsia: A case-control study in Southern SwedenPLOS ONE

Dear Dr.Malm,

Thank you for submitting your manuscript to PLOS ONE. After careful consideration, we feel that it has merit but does not fully meet PLOS ONE’s publication criteria as it currently stands. Therefore, we invite you to submit a revised version of the manuscript that addresses the points raised during the review process.

.I am sorry for the delay returning comments to you. Finding reviewers was difficult. I had hoped to find a second reviewer. .As I am in agreement with the one reviewer, I will send his/her comments. The reviewer was thoughtful. I therefore ask you to consider additions and changes based on this review.

Please submit your revised manuscript by 10/15/22. If you will need more time than this to complete your revisions, please reply to this message or contact the journal office at plosone@plos.org. Please include the following items when submitting your revised manuscript:

We look forward to receiving your revised manuscript.

Kind regards,

Barbara Wilson Engelhardt, MD

Academic Editor

PLOS ONE

Journal Requirements:

 “The work was supported by the Swedish Research Council for Sustainable Development (FORMAS, Dnr 217-00699), the ReproUnion, which was co-financed by the European Union, Interreg VÖKS, an ALF government funding (Dnr 2018-0075), and Lund University.” 

5. PLOS requires an ORCID iD for the corresponding author in Editorial Manager on papers submitted after December 6th, 2016. Please ensure that you have an ORCID iD and that it is validated in Editorial Manager. To do this, go to ‘Update my Information’ (in the upper left-hand corner of the main menu), and click on the Fetch/Validate link next to the ORCID field. This will take you to the ORCID site and allow you to create a new iD or authenticate a pre-existing iD in Editorial Manager. Please see the following video for instructions on linking an ORCID iD to your Editorial Manager account: https://www.youtube.com/watch?v=_xcclfuvtxQ.

7.Please review your reference list to ensure that it is complete and correct. If you have cited papers that have been retracted, please include the rationale for doing so in the manuscript text, or remove these references and replace them with relevant current references. Any changes to the reference list should be mentioned in the rebuttal letter that accompanies your revised manuscript. If you need to cite a retracted article, indicate the article’s retracted status in the References list and also include a citation and full reference for the retraction notice.

Reviewers' comments:

Reviewer's Responses to Questions

**Comments to the Author**

1. Is the manuscript technically sound, and do the data support the conclusions?

Reviewer #1: Yes

2. Has the statistical analysis been performed appropriately and rigorously? 

Reviewer #1: Yes

3. Have the authors made all data underlying the findings in their manuscript fully available?

Reviewer #1: Yes

4. Is the manuscript presented in an intelligible fashion and written in standard English?

Reviewer #1: Yes

5. Review Comments to the Author

Reviewer #1: To the authors:

Abstract:

Background--- why look at vitamin D in the first place?

Methods- well written. Were there any a priori hypotheses? Why was 34 weeks used as the time cut-point?

Results- clear

Conclusions- some sentence about the clinical importance of the results and the background of asking the question would be helpful.

Manuscript:

Background: this is clear. How do large population studies address vitamin D and its association with pre-eclampsia. Would a randomized trial be better at answering this question?

Methods: What is the ultrasound database? Is it a standard one- does it have citations associated with it?

Why perform logistic regression. Were there too many variables included in the analyses?

Results:

These are clear.

Discussion:

More discussion on the potential mechanisms is needed.

Why not ask for a clinical trial to answer the question. So many epi studies lead to false insights

6. PLOS authors have the option to publish the peer review history of their article (what does this mean?). If published, this will include your full peer review and any attached files.

Reviewer #1: No

---

## [Author Response · Author response to Decision Letter 0]

12 Oct 2022

Our response to the comments from editor and reviewer

1. Please provide additional details regarding participant consent. In the ethics statement in the Methods and online submission information, please ensure that you have specified (1) whether consent was informed and (2) what type you obtained (for instance, written or verbal, and if verbal, how it was documented and witnessed). If your study included minors, state whether you obtained consent from parents or guardians. If the need for consent was waived by the ethics committee, please include this information.

Comment: A sentence has been added in the under Materials and Methods: The ethical approval included that a specific consent for the study was not needed.

 “The work was supported by the Swedish Research Council for Sustainable Development (FORMAS, Dnr 217-00699), the ReproUnion, which was co-financed by the European Union, Interreg VÖKS, an ALF government funding (Dnr 2018-0075), and Lund University.” 

Comment: The following text has been added under Funding: The funders had no role in study design, data collection and analysis, decision to publish, or preparation of the manuscript. There was no additional external funding received for this study. 

Comment: Due to the personal data collected and potentially identifying information contained within the data, data are available upon request. Ethical approval by The Swedish Ethical Review Authority may be necessary, and requests for data may be sent to LUPOP - Lund University Population Research Platform to the email address: lupop@ed.lu.se.

l

4. PLOS requires an ORCID iD for the corresponding author in Editorial Manager on papers submitted after December 6th, 2016. Please ensure that you have an ORCID iD and that it is validated in Editorial Manager. To do this, go to ‘Update my Information’ (in the upper left-hand corner of the main menu), and click on the Fetch/Validate link next to the ORCID field. This will take you to the ORCID site and allow you to create a new iD or authenticate a pre-existing iD in Editorial Manager. Please see the following video for instructions on linking an ORCID iD to your Editorial Manager account: https://www.youtube.com/watch?v=_xcclfuvtxQ.

Comment: It is done. 

Comment: The sentence ‘data not shown’ is removed from the manuscript. We agree that it is redundant.

6.Please review your reference list to ensure that it is complete and correct. If you have cited papers that have been retracted, please include the rationale for doing so in the manuscript text, or remove these references and replace them with relevant current references. Any changes to the reference list should be mentioned in the rebuttal letter that accompanies your revised manuscript. If you need to cite a retracted article, indicate the article’s retracted status in the References list and also include a citation and full reference for the retraction notice.

Comment: Three more references have been added to the list. 

Our response to the comments from the reviewer:

1. Abstract:

Background--- why look at vitamin D in the first place?

Answer: A sentence justifying the study has been added in the abstract. 

2. Methods:

a. Were there any a priori hypotheses?

Answer: In the Abstract it is stated “…low serum vitamin D concentrations in early pregnancy and the risk of developing preeclampsia”. 

In addition, in the Background it is described in more detail “recent studies have suggested that low vitamin D levels in pregnancy are associated with increased risk of preeclampsia. Vitamin D may play a role in early placental development, thus there might be a preventive role for vitamin D supplementation. To clarify the link between vitamin D status and preeclampsia, large population-based studies, with high participation rate as well as valid data regarding pregnancy outcomes and potential confounders are needed. In the present study, the association between vitamin D levels and the risk of developing preeclampsia was investigated in a population-based study by combining comprehensive information from registers and analyzing the concentrations of vitamin D in maternal serum collected in early pregnancy before manifestation of preeclampsia.”

b. Why was 34 weeks used as the time cut-point? 

Answer: Due to limited numbers of words in the abstract this information is more described in the background: The definition for early-onset or ‘placental’ preeclampsia is when it occurs before 34 weeks whereas late-onset or ‘maternal’ preeclampsia occurs after 34 weeks. 

3. Conclusions - some sentence about the clinical importance of the results and the background of asking the question would be helpful.

Answer: The following sentence has been added: Preeclampsia is not a homogenous condition and more studies are needed before vitamin D supplementation during pregnancy can be recommended.

4. Manuscript:

Background: How do large population studies address vitamin D and its association with pre-eclampsia. Would a randomized trial be better at answering this question?

Answer: The results stress the importance that future studies, epidemiological studies as well as randomized trials, should not treat preeclampsia as a homogeneous pregnancy complication.

5. Methods: What is the ultrasound database? Is it a standard one- does it have citations associated with it?

Answer: Two new references regarding the ultrasound database have been added to the reference list. 

Why perform logistic regression. Were there too many variables included in the analyses?

Answer: Logistic regression is the standard method to use in case-control studies. In this study we have chosen to investigate and present both unadjusted and adjusted analyses. We agree with the reviewer that some of the adjusted analyses where we included relatively few individuals, has to be interpreted with some caution which we have tried to do. 

Discussion:

More discussion on the potential mechanisms is needed.

Answer: We agree with the reviewer that there are room for a more detailed discussion about potential mechanisms and have accordingly added the following text in the manuscript: “The protective effect of vitamin D on placental vascularization and angiogenesis takes places during placentation in early pregnancy [27]. It is therefore likely important to have adequate antenatal levels of vitamin D plasma levels. The mechanism behind the association between vitamin D and placenta function has not been fully elucidated. Calcitriol has been shown to regulate immune response and maintain immune homeostasis and thereby prevent placental vasoconstriction and ultimately preeclampsia. Also effects on endothelial and smooth cell proliferation and blood pressure regulation have been suggested to be involved in the interaction between vitamin D and cells in the placenta, possibly via vascular endothelial growth factor (VEGF)-mediated effects on angiogenesis.” 

Why not ask for a clinical trial to answer the question. So many epi studies lead to false insights

Answer: A clinical trial would of course been desirable with more participants in each group. This is now mentioned in the conclusion.

6. PLOS authors have the option to publish the peer review history of their article (what does this mean?). If published, this will include your full peer review and any attached files.

Do you want your identity to be public for this peer review? For information about this choice, including consent withdrawal, please see our Privacy Policy.

Reviewer #1: No

---

## [Decision Letter · Decision Letter 1]

19 Jan 2023

Maternal serum vitamin D level in early pregnancy and risk for preeclampsia: A case-control study in Southern Sweden

PONE-D-22-04216R1

Dear Dr. Malm,

We’re pleased to inform you that your manuscript has been judged scientifically suitable for publication and will be formally accepted for publication once it meets all outstanding technical requirements.

Kind regards,

Barbara Wilson Engelhardt, MD

Academic Editor

PLOS ONE

Additional Editor Comments (optional):

Dear Dr. Malm,

Your manuscript is now accepted for publication.

Please make sure your publication is entirely in accordance with PLOS procedure.

All the best,

Barbara Engelhardt MD, academic editor

Reviewers' comments:

Reviewer's Responses to Questions

**Comments to the Author**

1. If the authors have adequately addressed your comments raised in a previous round of review and you feel that this manuscript is now acceptable for publication, you may indicate that here to bypass the “Comments to the Author” section, enter your conflict of interest statement in the “Confidential to Editor” section, and submit your "Accept" recommendation.

Reviewer #1: All comments have been addressed

2. Is the manuscript technically sound, and do the data support the conclusions?

Reviewer #1: Yes

3. Has the statistical analysis been performed appropriately and rigorously? 

Reviewer #1: Yes

4. Have the authors made all data underlying the findings in their manuscript fully available?

Reviewer #1: Yes

5. Is the manuscript presented in an intelligible fashion and written in standard English?

Reviewer #1: Yes

6. Review Comments to the Author

Reviewer #1: This is a nice revision. All concerns have been addressed. No new edits are needed.

The edits to the abstract and methods are on point.

The authors conclusions are balanced and reflect the text.

7. PLOS authors have the option to publish the peer review history of their article (what does this mean?). If published, this will include your full peer review and any attached files.

Reviewer #1: No

---

## [Editor Report · Acceptance letter]

30 Jan 2023

PONE-D-22-04216R1 

*Maternal serum vitamin D level in early pregnancy and risk for preeclampsia: A case-control study in Southern Sweden*

Dear Dr. Malm:

I'm pleased to inform you that your manuscript has been deemed suitable for publication in PLOS ONE. Congratulations! Your manuscript is now with our production department. 

Kind regards, 

on behalf of

Dr. Barbara Wilson Engelhardt 

Academic Editor

PLOS ONE